# Performance Analysis of a Reduced Form-Factor High Accuracy Three-Axis Teslameter

**Johann Cassar [1],\*** , **Andrew Sammut [1]** , **Nicholas Sammut [2]** , **Marco Calvi [3]** , **Sasa Spasic [4]** and **Dragana Popovic Renella [4]**

[1]   Faculty of Engineering, University of Malta, MSD 2080 Msida, Malta; andrew.sammut@um.edu.mt
[2]   Faculty of ICT, University of Malta, MSD 2080 Msida, Malta; nicholas.sammut@um.edu.mt
[3]   Photon Science Division of the Paul Scherrer Institute, 5232 Villigen PSI, Switzerland; marco.calvi@psi.ch
[4]   SENIS, Hertizentrum 11, 6300 Zug, Switzerland; spasic@senis.ch (S.S.); dragana@senis.ch (D.P.R.)
\*   Correspondence: johann.cassar@um.edu.mt; Tel.: +356-2340-3339

**Abstract:** In the framework of the SwissFEL project at the Paul Scherrer Institute (PSI), a Hall probe bench is being developed for the high-precision magnetic characterization of the insertion devices for the ATHOS soft X-ray beamline. For this purpose, a novel three-axis teslameter has been developed, which will be placed between the undulator and its outer shell in a very limited volumetric space of $150 \times 50 \times 45$ mm. Together with a **SENIS® 3-axis** Hall probe at the center of the cross sectional area of the undulator, the setup will traverse along the undulator length on a specifically designed rig with minimal vibrations. This teslameter has all the analog signal conditioning circuitry for the Hall probe and also has on board 24-bit digitization. The instrument also handles an interface to a linear absolute encoder. The old instrumentation used only had analog signal conditioning circuitry whilst digitization was done off board. The new instrument also provides a very accurate magnetic field map in the μT range with simultaneous readings from the position encoder at an accuracy of ±3 μm. In this paper, a series of tests are described, which were performed at PSI in order to establish the measuring precision and repeatability of the instrument.

**Keywords:** analog-to-digital conversion; ATHOS soft X-ray beamline; Hall probe; three-axis teslameter; undulator

## 1. Introduction

Whilst various technologies are available for magnetic field measurement, Hall probes are commonly used for magnetic field map measurements when characterizing insertion devices. Insertion devices, also known as undulators, are powerful generators of synchrotron radiation in storage rings [1].

The scope of obtaining a field map tied down to a coordinate system across the whole undulator length enables the detection of kicks and imperfections in magnet pole positions across the undulator.

One of the undulator lines in SwissFEL at the Paul Scherrer Institute (PSI) is ATHOS, which is capable of covering the entire soft X-ray range from about 200 eV to 2 keV on the fundamental harmonic with full polarization control [2]. Each undulator module is 4 m long with a period of 40 mm and a physical magnetic gap in the range of 6.5 to 24 mm.

The characterization of the individual periods of one undulator segment is performed using a 3-axes Hall probe, which is moved longitudinally along the laser line. $B(z)$ is measured along the undulator length. The trajectory angle $\varphi$ given by Equation (1) and the offset $x$ given by Equation (2)

are calculated locally at every magnet period and are corrected by vertical adjustment of the keeper support and the horizontal adjustment of the pole [2].

$$\varphi = \int B{\cdot}dz \tag{1}$$

$$x = \int \int B{\cdot}dz^2 \tag{2}$$

For this purpose, a novel three axis teslameter has been developed that is interfaced to a SENIS type S Hall probe [3] for the high fidelity characterization of the new line of the ATHOS undulators.

## 2. Architecture of the Electronic Circuitry

The developed instrument comprises interfacing circuitry to a 3-axes Hall probe using the spinning current modulation technique [4–6] explained in detail in Section 3 and further amplified in [7]. Each axis of the Hall probe is biased with a very high precision and temperature independent 2.5 mA current source. As the Hall probe has an on-die PT100 sensor, this is also interfaced to the instrument using the four-wire configuration, so that lead wire resistance does not affect the true voltage temperature readout. In order to minimize any self-heating effects of the PT100, the bias current used is just 250 µA.

The four analog differential voltages are all amplified, demodulated, and low pass filtered to a 500 Hz bandwidth. This is performed by a 3rd order low pass fully differential Butterworth filter on each channel, which serves as an antialiasing filter for the ADC and the internal digital sinc$^3$ filter of the ADC. The antialiasing filter was designed with a bandwidth of 500 Hz, providing a ripple free response and no attenuation in the passband that covers the full frequency response of the Hall probe.

All the differential signal paths are length matched and routed parallel to each other to optimize CMRR (common mode rejection ratio). Also, each pair of the differential tracks are routed separately to the other pairs with a copper pour ground area in between in order to minimize crosstalk.

The four channel simultaneous sampling delta-sigma analog-to-digital converter [8] digitizes the four analog signals with 24-bit resolution. Oversampling techniques implemented through the delta-sigma architecture of the ADC enable the differential analog input voltage to be sampled at an effective frequency of 4.096 MHz from the delta-sigma modulator, as shown in Figure 1. The modulator then converts the analog input signal into a high-speed, pulse-wave representation. Further details of the implementation can be found in [7].

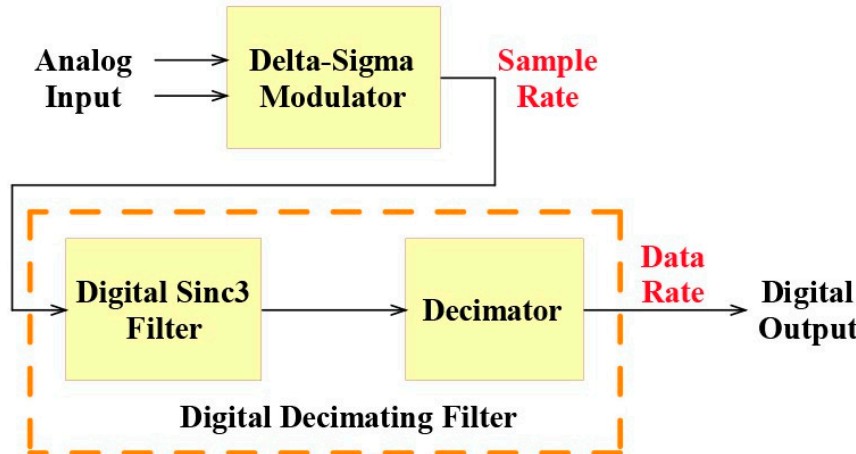

**Figure 1.** Internal block diagram of the ADC showing the core functions.

The third order sinc filter on each channel of the ADC works in the digital domain as data that is supplied to the filter from the modulator at the rate of $f_{MOD}$. A detailed phase analysis and magnitude analysis are presented in Section 6 that pertain to the frequency response of the sinc$^3$ filter. The digital

sinc$^3$ filter still operates at the modulator sampling rate for the decimator to be able to reduce the digital signal's output rate to the desired Nyquist frequency according to the output data rate. The decimating function works by accumulating and averaging together groups of 24-bit data. In this way, the actual output data rate is decimated down in the kHz range.

The noise performance of the ADS131A04 is best described in Table 1 for the output data rates of 1, 2, 4, and 8 kHz. This provides the theoretical noise figures, as from [8], in the functioning mode that the ADC is working in, whereby the analog supply voltage is ±2.5 V and the external reference voltage is 4.096 V. The effective number of bits and the RMS (root mean square) noise voltage figures are obtained with the analog inputs shorted together and by taking an average of multiple readings across all channels. Since full dynamic range covers ±2 T obtained through calibration and hardware amplification gain, noise figures are also presented in µT.

**Table 1.** Noise performance figures of the delta-sigma ADC for the different output data rates.

| $f_{DATA}$ at 4.096 MHz $f_{MOD}$/kHz | Effective Resolution/bits | $\mu V_{rms}$ | $\mu T_{rms}$ |
|:---:|:---:|:---:|:---:|
| 1 | 22.19 | 1.66 | 0.83 |
| 2 | 21.68 | 2.38 | 1.19 |
| 4 | 21.18 | 3.37 | 1.68 |
| 8 | 21.03 | 3.72 | 1.86 |

The dual core TMS320F28379D C2000 series Delfino microcontroller [9] is a powerful 32-bit floating point microcontroller.

The natural choice of communication and data transfer with the Raspberry Pi platform controlling the operation of the measurement rig is implemented through a USB 2.0 link. This USB interface serves both for the instrument to receive direct commands from the Raspberry Pi, such as initiation or termination of measurements, and also to transfer the acquired data during measurement from the on-board 128 MB SDRAM (single data rate random access memory). The size of the SDRAM has been based on the notion that since the ATHOS undulators are 4 m in length, and axis traversal is performed at a minimum of 10 mm/s, the maximum acquired data size at an output data rate of 8 kHz for a single back-and-forth mapping of the undulator will fit within 128 MB. Each reading is 20-bytes long, consisting of the X-axis, Y-axis, and Z-axis magnetic fields, temperature, and encoder position reading. After a complete traversal of the undulator axis, the acquired raw data is computed to calibrated data and then stored on the micro SD card or else transmitted via USB to the Raspberry Pi.

The instrument also supports an RS-485 interface to a Heidenhain linear absolute encoder. Communication to the LIC 4117 Heidenhain encoder is done using the EnDat 2.2 protocol, which is a digital bidirectional interface standard for position and rotary encoders [10]. An accuracy of ±3 µm with a measuring step of 10 nm is provided by the encoder. The magnetic field readings and the physical position reading are synchronized. The time lag between the falling edge of the ADC interrupt signal indicating that data is readily available to be clocked out and the start of the encoder polling transmission command amounts to 8.4 µs, being equivalent to a total of 1680 clock cycles for the microcontroller to handle the request.

The instrument architecture is presented in Figure 2, which shows the different blocks of the instrument circuitry both for the analog and the digital domain as explained. This developed architecture provides a tailored solution for this application in mapping the magnetic field across the undulator length with respect to position. In this way, better performance in the synchronization timing and noise performance is achieved.

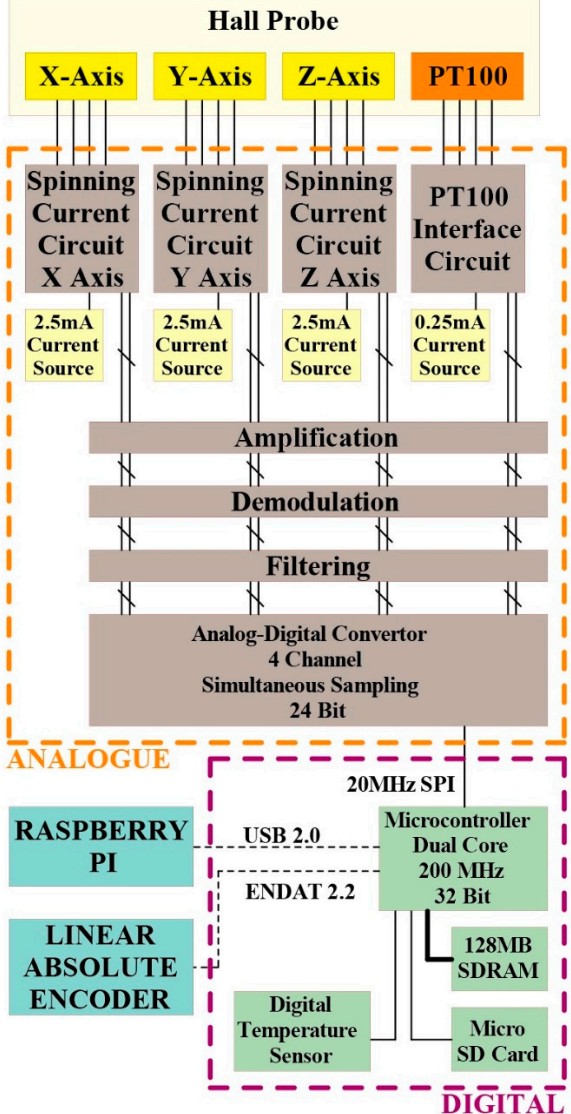

**Figure 2.** Block diagram of the instrument comprising the analog domain and the digital domain sections each composed of different blocks. The instrument also supports an external interface to the Raspberry Pi and the linear absolute encoder.

The instrument circuit board is an 8-layer PCB (printed circuit board). The PCB is 144 mm long and 44 mm wide. The PCB layer stack-up incorporates four signal layers, two internal ground planes, and two internal power supply planes, which make it possible to condense such complex circuitry in a very tight physical space. The two signal layers in the middle, sandwiched between the two ground planes, provide excellent noise immunity for the sensitive and noise-prone analog signal tracks. The layout of the PCB, as shown in Figure 3, incorporates all the circuitry blocks presented in Figure 2 with all the analog circuitry partitioned on the left-hand side of the board away from the digital circuitry to minimize any digital switching noise from entering the analog section. Also, the analog and the digital ground planes are connected via a high impedance star point beneath the ADC.

Figure 3 is a photo of the complete instrument in a tailor-designed and manufactured 1.6 mm thick grey powder-coated aluminum enclosure. The PCB is screwed to 6 mm high standoffs. A cooling fan is mounted on the top cover of the enclosure and extracts the heat generated by the microcontroller to minimize any temperature hotspots and to keep the temperature as stable as possible.

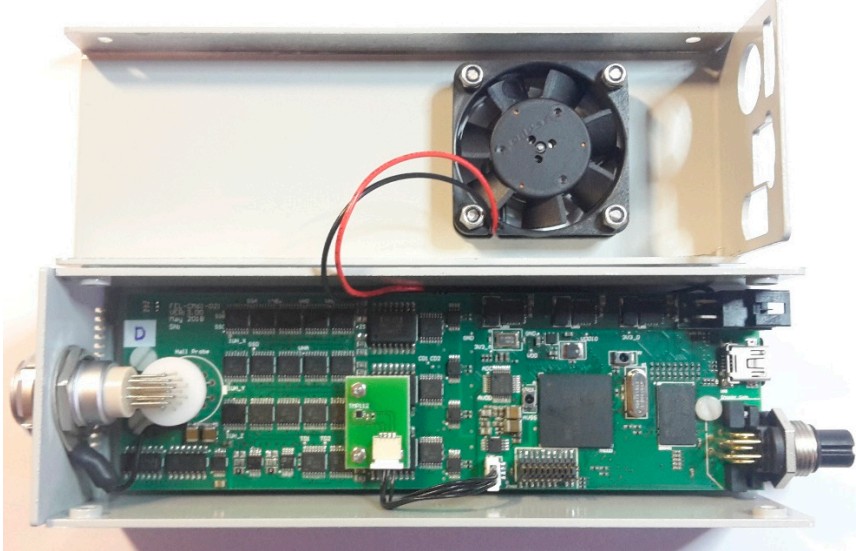

**Figure 3.** Photo of the complete instrumentation board inside the aluminum enclosure. The Hall probe connector is located on the narrow left-hand side of the board with all digital and supply connectors on the opposite side.

## 3. Spinning Current Modulation Technique

As explained in [5], the signal $V_{out}$ at the output of the analog chain is composed of different contributions. This can be denoted by Equation (3):

$$V_{out} = G\left(V_H + V_{Hall\ Offset} + V_{PA\ Offset} + e_{th} + V_{ind} + V_{Hall\ LF} + V_{PA\ LF} + v_{th}\right) \tag{3}$$

where $G$ is the overall electronic gain, $V_H$ is the true Hall voltage, $V_{Hall\ Offset}$ and $V_{PA\ Offset}$ are the raw offsets of the Hall plate and the amplification stage, respectively. $e_{th}$ is the sum of the thermoelectric voltages in the output circuit between the sense outputs and the pre-amplifier (PA) input. $V_{ind}$ is the pickup voltage in the output circuit. The terms $e_{th}$ and $V_{ind}$ come into effect due to the physical interconnections between the Hall probe output and the amplification stage. $V_{Hall\ LF}$ and $V_{PA\ LF}$ are the low frequency noise of the Hall plate and the PA, respectively. The white noise $v_{th}$ is contributed by the Hall plate and the PA stage. Therefore, DC (direct current) bias excitation of the Hall probe will result in an output voltage far from the true Hall voltage as all these parasitic components add up.

Proper compensation of offsets and drifts is attained using the spinning current modulation technique [4–6]. This technique allows dynamic cancellation of the offset and low frequency noise. The bias and sense terminals of the Hall plate are periodically swapped. A Hall device, as depicted in Figure 4, is commonly modelled as a four terminal resistive Wheatstone bridge [1], whereby the inherent offset is linked to an imbalance in one of the resistive legs. This offset that appears at the output even in the absence of a magnetic field, finds its origins due to a structural asymmetry of the active part and also alignment errors of the sense contacts, one being further upstream and the other downstream with respect to the bias current [1].

One arbitrarily defines the orientation given by PH (phase) 1 as shown in Figure 4, to produce a Hall voltage in the given direction with an offset voltage opposing the Hall voltage. Bias current rotation is implemented through PH 2 to PH 4, whereby the Hall voltage rotates in the same direction as the bias current, indicated by the green arrows in Figure 4. Hall voltage polarity depends only on the current direction. As the offset voltage remains static in the Hall sensor throughout the rotation, its polarity is different than that of the Hall voltage. Amplification of the Hall voltage output and the offset together with their inversions is done to obtain the full dynamic range and keep the signal chain fully differential.

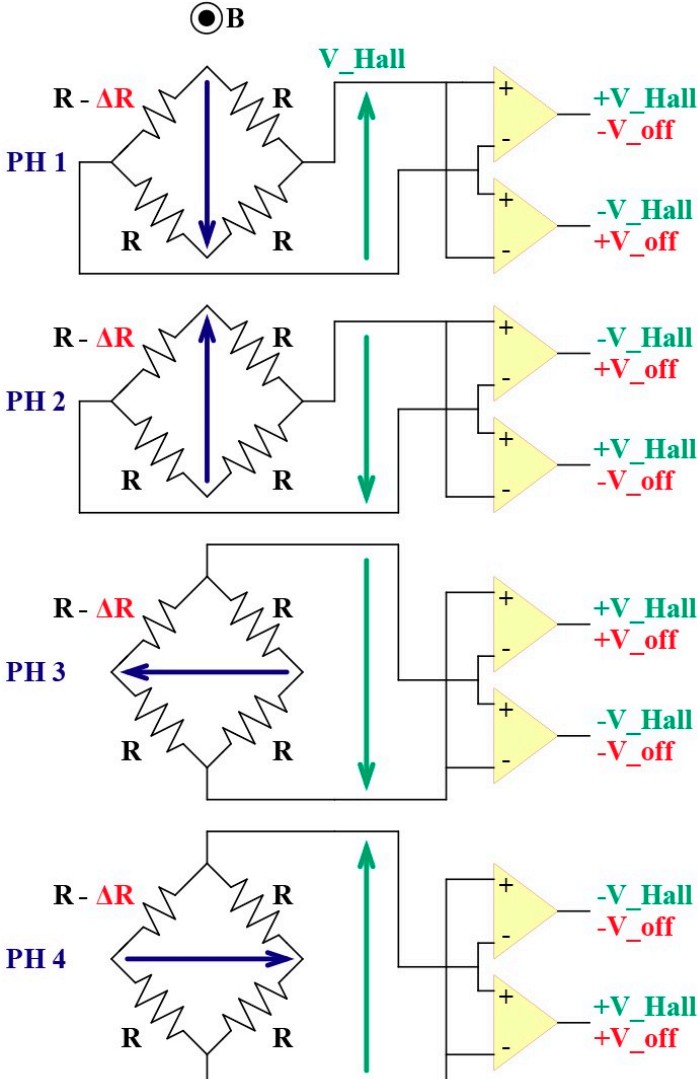

**Figure 4.** Four phase spinning current sequence. The inherent resistance imbalance in the Hall probe produces an offset voltage in which polarity does not follow that of the true Hall voltage, which depends only on the current direction. Therefore, this can be filtered out.

For the given spinning sequence depicted in Figure 4, the digital demodulation key needed for recovering the pure Hall signal is given by the signs of the Hall signal after modulation. The two top graphs in Figure 5 depict the output response of the amplification stage after spinning is applied.

Demodulation is performed using an additional set of analog switches, whose switching configuration determines the signal output. The demodulation key, outlined in Figure 5 in red and also given in Table 2, provides the correct differential voltage with only a polarity change in the offset that can be filtered out. The offset of the Hall device and the PA offset can be extracted instead of the true Hall voltage by choosing other demodulation keys as suggested in Tables 3 and 4 and further explained in [5].

The spinning of the current through the Hall probe is controlled using the first set of synchronized PWM (pulse width modulation) signals as depicted in Figure 6. One whole period comprising of four phases takes 128 µs (7.8 kHz). The spinning sequence is chosen to be at a much higher frequency than the bandwidth of the Hall probe. This allows proper filtering of the switching noise and its related harmonics without deteriorating the bandwidth of interest.

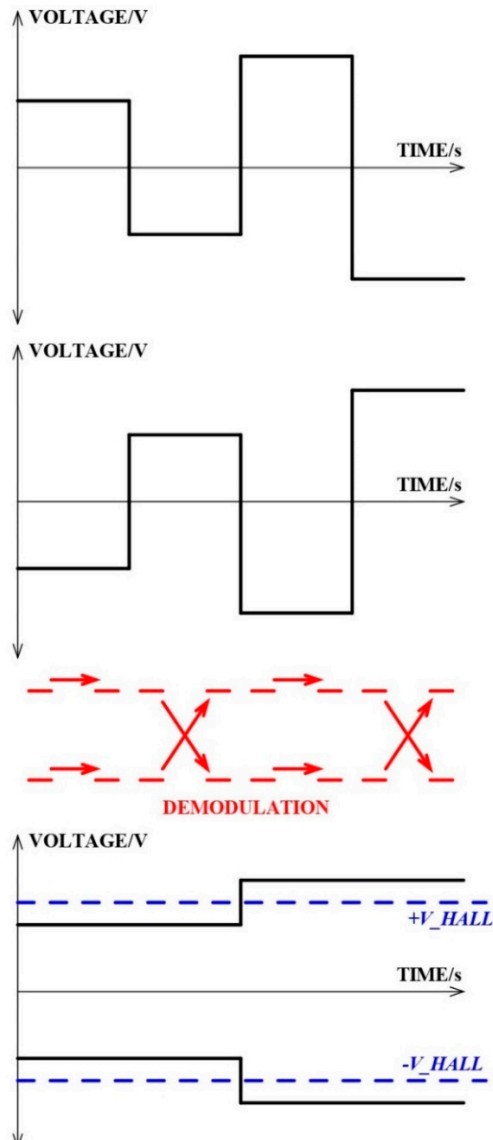

**Figure 5.** The two top graphs show the output response of the amplification stage with the offset superimposed on the true Hall voltage. The demodulation key shown in red applied on the top graphs produces the differential true Hall voltage output upon low pass filtering.

**Table 2.** Demodulation key applied on the amplified spinning output for the extraction of the true Hall voltage. PA: pre-amplifier.

| PH | After Spinning stage at PA input | | | Demod Sign | After Demod |
|----|-------------|-------------|-----------|------------|-------------|
|    | Hall Signal | Hall Offset | PA Offset |            | Hall Signal |
| 1  | +           | +           | +         | +          | +           |
| 2  | -           | -           | +         | -          | +           |
| 3  | +           | -           | +         | +          | +           |
| 4  | -           | +           | +         | -          | +           |

Voltage readout from the Hall probe requires a change in the connection to the amplification stage in PH 3 and PH 4, as shown in Figure 4. This entails a voltage readout spinning circuit, controlled using two PWM signals, depicted in blue in Figure 6.

**Table 3.** Demodulation key applied on the amplified spinning output for the extraction of the Hall offset.

| PH | After Spinning stage at PA input | | | Demod Sign | After Demod |
| --- | --- | --- | --- | --- | --- |
| | Hall Signal | Hall Offset | PA Offset | | Hall Offset |
| 1 | + | + | + | + | + |
| 2 | - | - | + | - | + |
| 3 | + | - | + | - | + |
| 4 | - | + | + | + | + |

**Table 4.** Demodulation key applied on the amplified spinning output for the extraction of the amplification stage offset.

| PH | After Spinning stage at PA input | | | Demod Sign | After Demod |
| --- | --- | --- | --- | --- | --- |
| | Hall Signal | Hall Offset | PA Offset | | PA Offset |
| 1 | + | + | + | + | + |
| 2 | - | - | + | + | + |
| 3 | + | - | + | + | + |
| 4 | - | + | + | + | + |

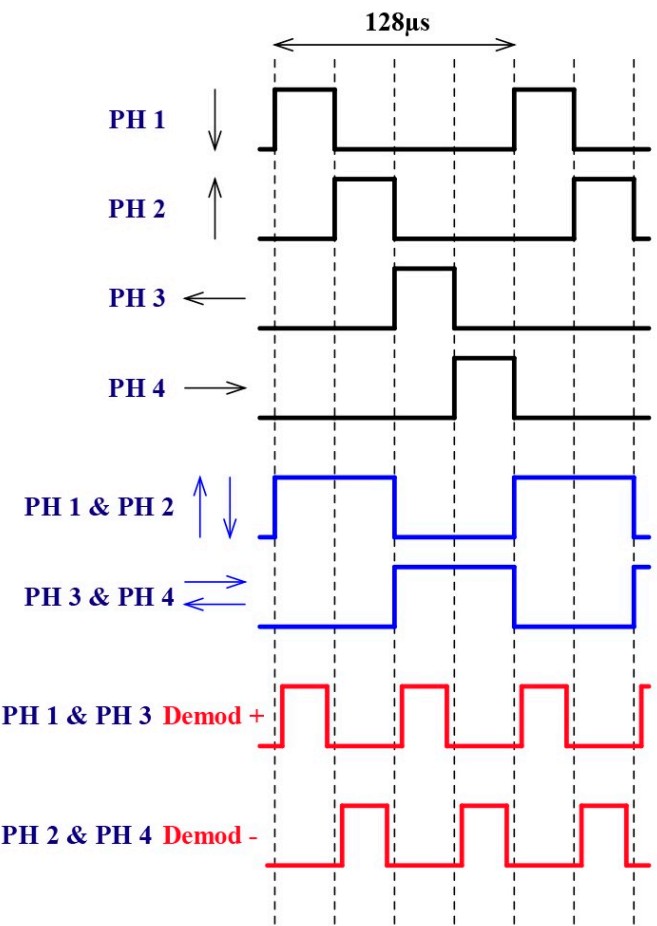

**Figure 6.** The set of eight signals required for proper spinning application. The top four PWM (pulse width modulation) signals control the current direction through the Hall probe with a period of 128 μs. The blue signals control the voltage readout polarity and the bottom red signals determine the demodulation key applied.

## 4. Instrument Calibration

Performance measurement of the instrument is based on fully calibrated data. As the calibration of the instrument is quite a complex and long process, an overview is given here that revolves around the calibration block diagram presented in Figure 7. The instrument is calibrated over the ±2 T range for all three axes. The highly nonlinear voltage output of the Hall probe with respect to the true magnetic field applied is calibrated using a 5th order polynomial. This is performed by exposing the Hall probe to the whole ±2 T range in 50 mT steps and the digitized voltage output from the instrument together with actual NMR readings are recorded simultaneously. The residual percentage standard deviation error decreases from 1.17% down to 0.004% upon non-linearity calibration. This calibration step also compensates for the constant offset registered by the Hall probe at an ambient temperature of 24 °C, which is nulled.

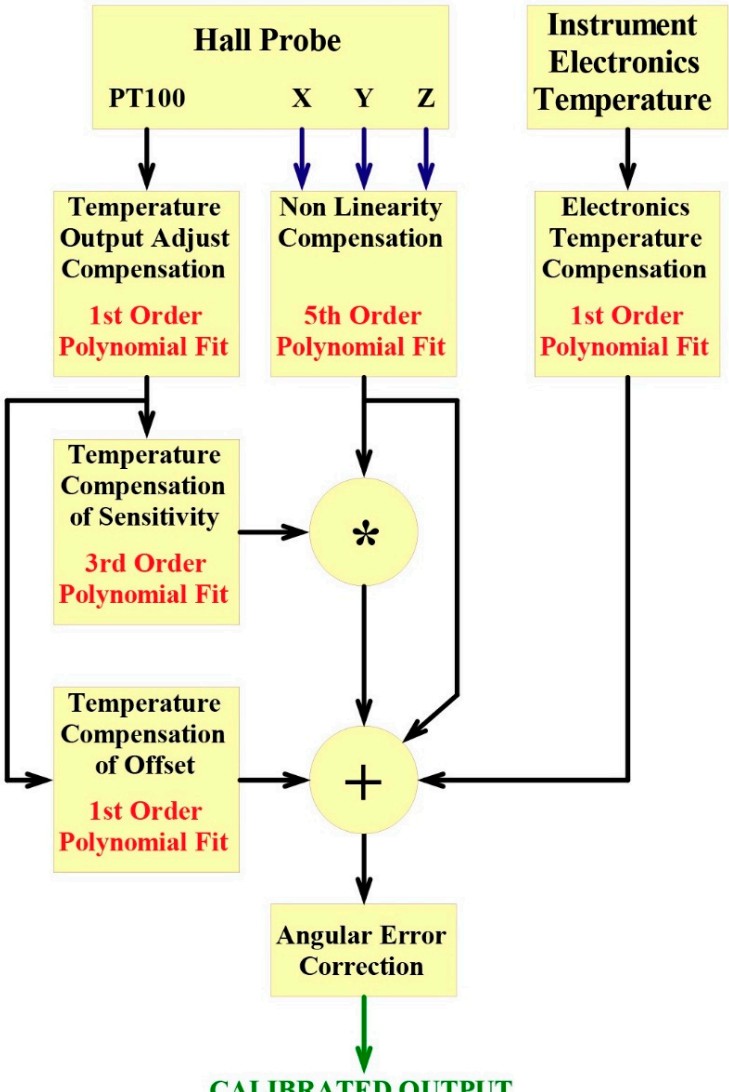

**Figure 7.** Calibration block diagram showing all necessary steps undertaken for a full proper calibration of the setup.

As the Hall probe analog output is very temperature-dependent, temperature compensation of the offset and the sensitivity are necessary steps for a proper calibration. Therefore, as the Hall probe comprises an on-die PT100 sensor, the temperature of the Hall probe is read using the "Temperature Output Adjust Compensation" block, which translates the voltage across the PT100 sensor to an actual

temperature value. Since the PT100 is a platinum resistive element with a constant positive temperature coefficient, this relationship is a linear one. This temperature reading is directly used to model the change in the offset as the Hall probe is at zero gauss for the temperature range from 14 °C up to 34 °C. This is modelled using a first order polynomial where the difference in the offset registered between the 24 °C scenario and any other temperature is subtracted.

As the sensitivity of the Hall probe changes according to the temperature, this must be calibrated by performing temperature hysteresis sweeps for various plateau values across the full magnetic field dynamic range. The change in the sensitivity is modelled using a cubic polynomial fit, which results in a further reduction of the error after linearization.

Variation in the instruments' electronics temperature must also be calibrated since this predominantly affects the gain set by the external resistors in the amplification stage. Therefore, the instrument is placed in a temperature chamber whilst the Hall probe is kept at zero gauss in a highly stable ambient temperature environment. The additional offset introduced when the instrument is at ambient temperatures other than 24 °C is linearly compensated.

All the corrections are added to the signal at the point indicated by the summer in Figure 7 and the calibrated value is output. This calibration is performed individually for all three axes. However, as it is physically impossible for the three Hall sensor dies to be perfectly orthogonally oriented to each other, angular errors of about 1° must be calibrated. As explained in [11], this calibration entails placing the probe at three precisely known angular positions in the magnetic field with known components, and the Hall output voltages are read. For each sensitivity axis, a set of three linear equations with three unknowns are obtained, which are then solved. The sensitivity tensor with all nine components is found and applied as the last calibration step, which reduces the effective angular errors of the Hall probe to less than 0.1°, as stated in [11].

## 5. Instrument Performance Measurements

This section provides an overview of a series of tests carried out at PSI facilities on an AppleX undulator in order to measure and analyze the performance of the instrument. The instrument was placed on a moving test rig and the Hall probe was also mounted securely to the bench. As shown in Figure 8, the whole setup was programmed to move at a defined constant velocity and the Hall Probe was passed through a gap between the magnets array.

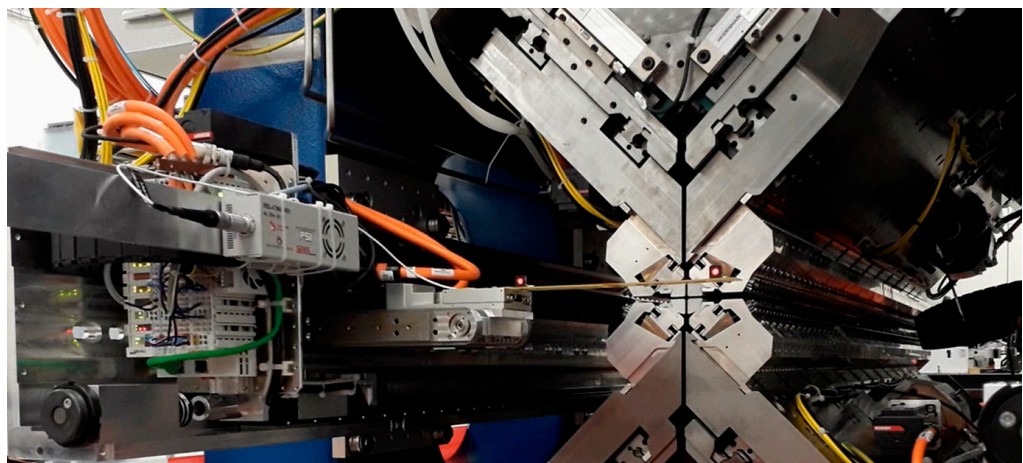

**Figure 8.** AppleX undulator prototype at Paul Scherrer Institute (PSI) facilities used for initial performance testing of the instrument.

### 5.1. Repeatability Analysis

The reproducibility of the instrumentation setup was tested over a 10 h period for 185 traversal runs of the AppleX undulator. Figure 9 shows the results obtained of the errors in the magnetic field

peak value taken with respect to the average. The margin of error was shown to be within ±0.5 G. This translates into an average percentage error of 0.005%, where the peak magnetic field of the undulator was ±1 T.

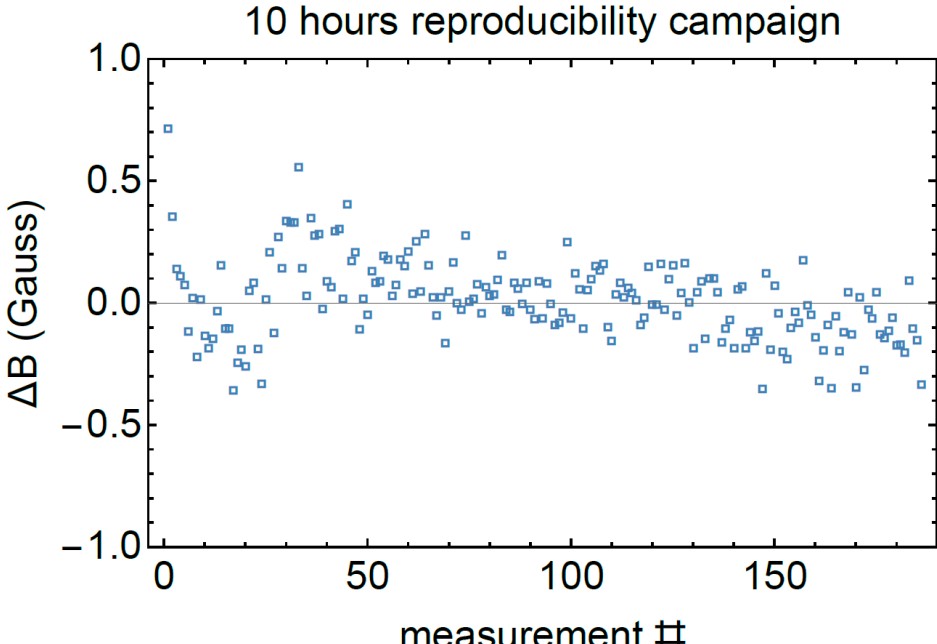

**Figure 9.** Repeatability performance plot showing a maximum deviation peak error of ±0.5 G over 185 undulator traversal runs.

*5.2. Noise Performance Analysis*

The noise performance analysis revolves around the 1/f noise spectrum and the broadband noise spectrum. The 1/f noise spectrum was obtained by limiting the cut-off frequency to 10 Hz, while white noise for the full bandwidth was measured in the broadband noise spectrum. Both the AC and the DC noise were measured by acquiring data over 10 s while the Hall probe was placed in a zero gauss chamber.

5.2.1. Offset Fluctuation and Drift (0.1–10 Hz)

The offset fluctuation and drift were measured in the frequency bandwidth from 0.1 to 10 Hz, as explained in [7]. Table 5 shows the results of the noise obtained before low pass filtering to a 10 Hz bandwidth.

**Table 5.** Noise performance figures of the calibrated output from the instrument at different output data rates.

| Before 10 Hz External LPF (low pass filter) | | |
|---|---|---|
| **Output Data Rate/kHz** | **1σ Error/μT** | **Peak-to-Peak Error/μT** |
| 1 | 1.7701 | 10.6207 |
| 2 | 2.3569 | 14.1414 |
| 4 | 2.6350 | 15.8105 |
| 8 | 3.1333 | 18.7999 |

By limiting the bandwidth to 10 Hz, the 1/f noise spectral density was computed using Equation (4). $f_L$ and $f_H$ denote, respectively, the low and high limits of the frequency range and $NSD_{1/f}$ denotes the noise spectral density at 1 Hz. This gives us the results as shown in Table 6.

$$V_{rmsB} \approx \left[ NSD_{1/f}^2 \cdot 1Hz \cdot ln \frac{f_H}{f_L} \right]^{\frac{1}{2}} \tag{4}$$

**Table 6.** Offset fluctuation and drift noise performance figures for the bandwidth of 0.1–10 Hz. $NSD_{1/f}$: noise spectral density at 1 Hz.

| | After 10 Hz External LPF | |
|---|---|---|
| **Output Data Rate/kHz** | **1σ Error/μT** | **$NSD_{1/f}$/μT/ $\sqrt{Hz}$** |
| 1 | 0.7829 | 0.3648 |
| 2 | 1.0945 | 0.5100 |
| 4 | 1.1608 | 0.5409 |
| 8 | 1.2664 | 0.5901 |

The noise in the frequency range from 0.1 to10 Hz is optimal at the output data rate of 1 kHz. The standard deviation of the noise is 0.78 μT. At the 8 kHz output data rate, the standard deviation of the noise increases to 1.26 μT.

### 5.2.2. Broadband Noise (10 Hz to $f_T$)

The white noise spectral density was computed using Equation (5). Results of the noise spectral density at each output data rate are given in Table 7.

$$V_{rmsB} \approx \left[ 1.22 \cdot NSD_W^2 \cdot f_T \right]^{1/2} \tag{5}$$

**Table 7.** Broadband noise performance figures for the bandwidth of 10–500 Hz.

| **Output Data Rate/kHz** | **1σ Error/μT** | **$NSD_W$/μT/ $\sqrt{Hz}$** |
|---|---|---|
| 1 | 1.567720 | 0.0634 |
| 2 | 2.056552 | 0.0832 |
| 4 | 2.395707 | 0.0969 |
| 8 | 2.864964 | 0.1159 |

### 5.2.3. Noise Performance Specifications Summary

A summary of the noise figures obtained for both DC and AC noise at all output data rates is presented in Table 8.

**Table 8.** Summary of all DC and AC noise figures.

| Offset Fluctuation and Drift (0.1–10 Hz)/$\mu T_{PP}$ | Output Data Rate/kHz |
|---|---|
| 4.69 | 1 |
| 6.56 | 2 |
| 6.96 | 4 |
| 7.59 | 8 |
| Noise Spectral Density @ f > 1 Hz ($NSD_1$)/$\mu T/Hz^{1/2}$ | Output Data Rate/kHz |
| 0.36 | 1 |
| 0.51 | 2 |
| 0.54 | 4 |
| 0.59 | 8 |

**Table 8.** *Cont.*

| Broad Band Noise (10 Hz to $f_T$)/$\mu T_{PP}$ | Output Data Rate/kHz |
|---|---|
| 9.36 | 1 |
| 12.3 | 2 |
| 14.34 | 4 |
| 17.16 | 8 |
| Noise Spectral Density @ f > 10 Hz (NSD$_W$)/$\mu$T/Hz$^{1/2}$ | Output Data Rate/kHz |
| 0.06 | 1 |
| 0.08 | 2 |
| 0.09 | 4 |
| 0.11 | 8 |

## 6. Instrument Frequency Analysis

### 6.1. Phase Analysis

As clearly explained in [7,12], the digital filter in the delta-sigma ADC with a sinc$^3$ architecture introduces a time delay between the analog input and digital output. A breakdown of the delay is presented in Table 2 in [7].

The transfer function of the sinc$^3$ filter is denoted by Equation (6):

$$H\left(e^{j\frac{f}{f_{MOD}}}\right) = \left[\frac{1}{D}x\frac{sin\left(D\frac{\pi f}{f_{MOD}}\right)}{sin\left(\frac{\pi f}{f_{MOD}}\right)}x\,e^{-j(D-1)\frac{\pi f}{f_{MOD}}}\right]^3 \tag{6}$$

where $f_{MOD}$ is the frequency of the modulator clock, which is 4.096 MHz in this case, and $D$ is the decimation rate. As group delay is defined as the number of samples in delay as a function of frequency, its mathematical definition is denoted by Equation (7).

$$Group\,Delay(\omega) = \frac{d\Phi(\omega)}{d\omega} \tag{7}$$

where $\Phi(\omega)$ is the phase response of the system and $\omega$ is the angular frequency measured in radians per sample.

In order to see the exact effects of the delays caused by the ADC and the improvement obtained after proper delay compensation, a series of tests were carried out on a phase matcher setup at PSI. The Hall probe was traversed at different velocities, ranging from 10 to 50 mm/s through a gap between a magnet array with a Mexican hat shaped magnetic field, as depicted in Figure 10. The output data rate frequency of the instrument was also changed accordingly throughout the experiment.

By taking the reference traversal reading to be at 1 kHz at 10 mm/s, interpolation techniques were used on the other traversal readings at the remaining velocities and output data rates to calculate the percentage errors in each case. The percentage error was taken with respect to the peak of the magnetic field during the traversal of the rig.

Figure 11 shows the scenario before any delay compensation was applied, where it is noted that the error magnitudes do not fall within the repeatability margin and so proper compensation for the group delays as shown in Table 9 needs to be applied. Figure 12 shows the scenario after delay compensation was applied for the same previous velocities. Table 9 summarizes the peak percentage errors obtained both before and after compensation.

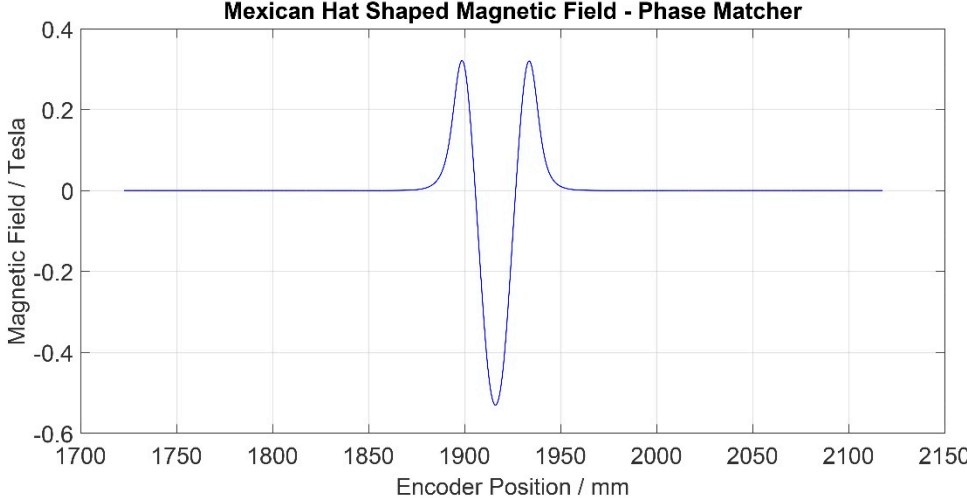

**Figure 10.** The response of the instrument output when traversing the magnet array with a Mexican hat shaped magnetic field with peak values of 0.3 and −0.55 T.

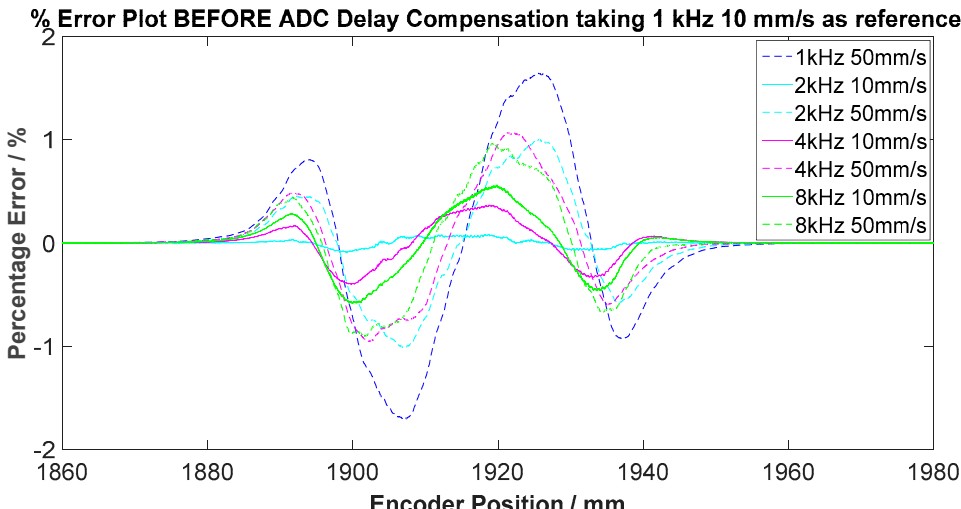

**Figure 11.** Percentage error plot obtained by applying interpolation on the 1 kHz 10 mm/s trajectory before delay compensation. As the group delay is a function of the output data rate (decimation rate D), the greatest delay was registered at the 1 kHz 50 mm/s case.

**Table 9.** Percentage errors comparison before and after delay compensation was applied.

| Output Data Rate/kHz | Velocity/mm/s | BEFORE Delay Comp % Peak Error | AFTER Delay Comp % Peak Error |
|---|---|---|---|
| 1 | 50 | 1.70675 | 0.2369 |
| 2 | 10 | 0.0904 | 0.2079 |
| 2 | 50 | 1.0142 | 0.4964 |
| 4 | 10 | 0.5379 | 0.4007 |
| 4 | 50 | 1.0730 | 0.9187 |
| 8 | 10 | 0.7772 | 0.5850 |
| 8 | 50 | 1.0463 | 1.0966 |

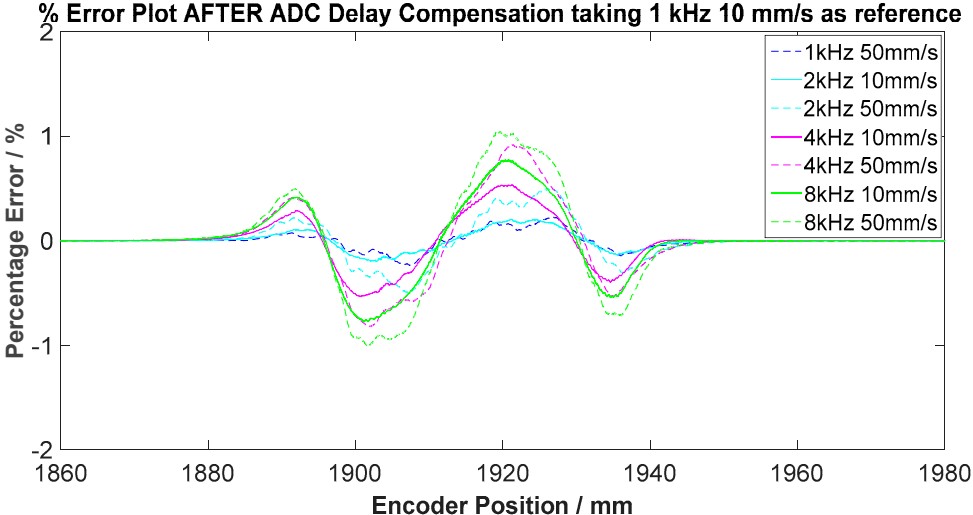

**Figure 12.** Percentage error plot obtained by applying interpolation on the 1 kHz 10 mm/s trajectory after delay compensation.

*6.2. Magnitude Analysis*

Figure 13 shows the magnitude responses for different output data rates of the filter, up to an input signal frequency of 500 Hz. Due to the nature of the transfer function of the $sinc^3$ filter, the filter has theoretically infinite attenuation at the output data rate and its multiples. It is to be noted that the bandwidth of the $sinc^3$ filter varies according to the output data rate set. The -3 dB point is represented graphically by the red horizontal line in Figure 13.

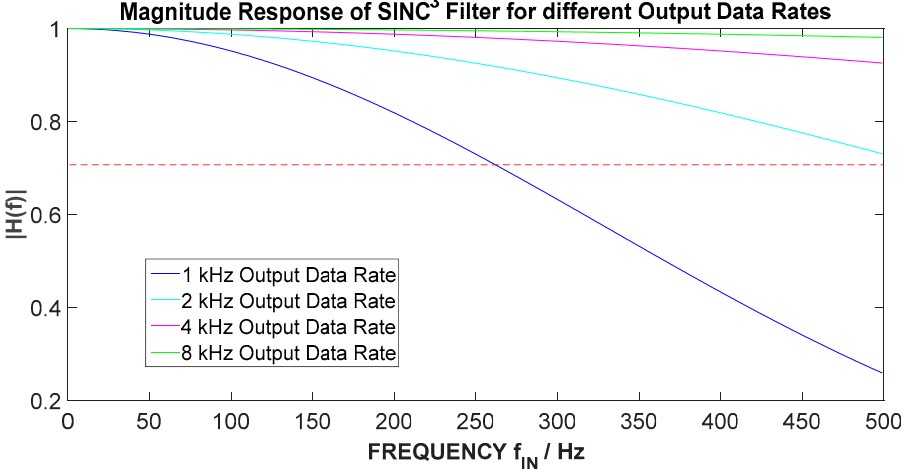

**Figure 13.** Graphical representation of the magnitude response of the $sinc^3$ filter for different output data rates.

Therefore, as each of the analog inputs to the $sinc^3$ filter are not purely DC, but contain additional frequency components in their spectrum, predominantly the switching spikes harmonics at 7.8 kHz and its multiples; these will be attenuated differently according to the output data rate frequency. These switching harmonics were primarily filtered by the 3rd order 500 Hz cut-off antialiasing Butterworth filter, which has a finite attenuation of −51 dB at 7.8 kHz, as shown in Figure 14.

Also, as the delta-sigma ADC is effectively a sampling system, all noise and signals above one half of $F_{MOD}$ alias back to the system bandwidth, as shown in Figure 15. This imposes a high order filter requirement in front of the delta-sigma ADC, as used in this case, rather than a simple first order RC low pass filter.

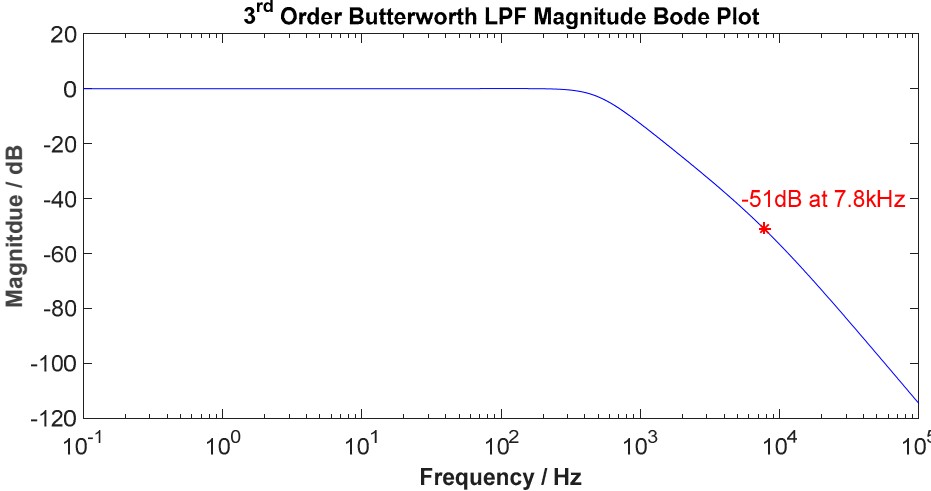

**Figure 14.** Magnitude Bode plot of the 3rd order Butterworth LPF showing an attenuation of -51 dB at 7.8 kHz.

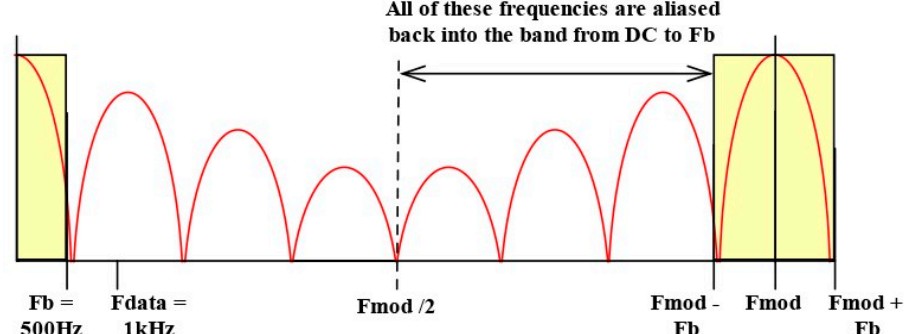

**Figure 15.** Aliasing of the upper band of frequencies from ($F_{MOD}$/2) to ($F_{MOD}$/2 – $F_b$) to the frequency bandwidth of interest from DC to $F_b$. This is problematic and so requires a high order antialiasing low pass filter before digitization.

Considering the first harmonic switching frequency component at 7.8 kHz, its alias frequency was predicted based on the sample rate. Equation (8) shows that the alias frequency is a function of the absolute value of the difference between the input signal frequency and the closest integer multiple of the sample rate:

$$f_a(N) = \left| f_{in} - Nf_s \right| \tag{8}$$

where $f_a$ is the alias frequency, $f_{in}$ is the input signal frequency, $f_s$ is the sample rate, and N is an integer greater than or equal to 0. Therefore, for the case of the 1 kHz sample rate:

$$f_a = \left| 7.8 \text{ kHz} - 8.0(1 \text{ kHz}) \right| = 200 \text{ Hz}. \tag{9}$$

As the aliasing frequency was 200 Hz, this frequency component was attenuated differently by the sinc$^3$ filter, as shown in Figure 13. Table 10 shows the attenuation factors at 200 Hz by the sinc$^3$ filter for the different sample rates.

Similarly, the second harmonic of the switching frequency occurring at 15.6 kHz, was aliased to 600 Hz. Therefore, the lower attenuation values at higher output data rates essentially translate into gain factors. This imposes the restriction that the calibration polynomial coefficients used for each output data rate have slightly different values in order to compensate this gain factor.

**Table 10.** Attenuation factors at 200 Hz by the sinc$^3$ filter.

| Output Data Rate/kHz | Sinc$^3$ Filter Transfer Attenuation Factor | Sinc$^3$ Filter Transfer Attenuation Factor/dB |
|---|---|---|
| 1 | 0.8186 | −1.7376 |
| 2 | 0.9516 | −0.4300 |
| 4 | 0.9877 | −0.1072 |
| 8 | 0.9969 | −0.0267 |

## 7. Discussion and Conclusions

The development and the performance of a novel three axes teslameter, which will be used in the characterization of the ATHOS beamline undulators, has been presented throughout this paper.

Repeatability performance tests yield an average percentage error of 0.005% over a magnetic field peak of ±1 T at an output data rate of 1 kHz.

Noise performance analysis shows that the best case DC offset fluctuation and drift has a standard deviation of 0.78 μT at a 1 kHz output data rate over a 10 Hz bandwidth. Worst case AC noise at a 500 Hz bandwidth amounts to a standard deviation of 2.05 μT at a 2 kHz output data rate.

Magnitude frequency analysis of the internal sinc$^3$ filter in the ADC shows a direct dependence of the bandwidth on the output data rate. An output data rate of 1 kHz was found to limit the bandwidth of the sinc$^3$ filter to 262 Hz, which does not cover the full Hall probe bandwidth. The best tradeoff in this scenario was the application of the 2 kHz output data rate when considering that the full bandwidth of interest was attained, and noise figures still remained within an acceptable range.

However, apart from the noise and bandwidth, another variable to be considered is the data transfer times of the acquired data to the on board micro SD card and via the USB interface after measurement time. As one minute of measurement time at an output data rate of 1 kHz yields 8 s and 20 s of transfer times for the USB and micro SD card, respectively, these transfer times will then double with the output data rate. This must also be considered in general if the total experimental time is an issue and is preferably minimized.

Calibration of the instrument at each output data rate was required, as the aliasing of the switching harmonics result in a magnitude difference of the output digitized signal due to the different attenuation factors at different output data rates.

Phase frequency analysis shows the occurrence of a group delay also dependent on the output data rate, which was constant for all frequency components of the analog input signal to the ADC due to the linear time invariant nature of the sinc$^3$ filter. This delay was compensated prior to applying the calibration algorithm.

This article has given a very detailed analysis of the performance of a novel three-axes teslameter. In comparison to other commercially available products [13], this instrument compares very well and offers advantages mainly in the integration of the analog-to-digital conversion and the spinning current analog readout circuit on the same board. An additional novelty is the integration of a digital interface to a Heidenhain linear absolute encoder, which provides synchronized position readings to magnetic field data. The novel three-axis teslameter is now commercially available under the name SENIS® MLNT-3D - 3-Axis Miniature Low Noise Digital Magnetic Transducer [14].

**Author Contributions:** Formal analysis, J.C.; Investigation, J.C.; Methodology, J.C.; Software, J.C.; Supervision, A.S. and N.S.; Validation, A.S., N.S., M.C., S.S., and D.P.R.; Writing—Original draft, J.C.; Writing—Review & editing, A.S., N.S., M.C.

**Funding:** This research received no external funding.

**Acknowledgments:** The authors would like to thank Mark Dalli for assisting in performing the tests at PSI facilities. The authors would like to thank R. Ganter, project leader of the ATHOS undulator beamline, and H-H. Braun, project leader of the SwissFEL accelerator, for their support throughout the entire project. The authors would like to thank Sasa Dimitrijevic, Marjan Blagojevic, and their team at Sentronis laboratory facilities at Nis Serbia for their support in the calibration process.

**Conflicts of Interest:** The authors declare no conflict of interest.

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
