# Peer review of "Performance Analysis of a Reduced Form-Factor High Accuracy Three-Axis Teslameter"

_electronics, doi:10.3390/electronics8111230_

Round 1
Reviewer 1 Report
The authors present test results of a Hall probe bench developed for a high-precision magnetic measurements. They describe in details a three-axis teslameter, in particular, important methods to compensate the effects of heating (temperature compensation) and of the delay of the output results. The paper is well organized.
As minor revisions I would suggest that authors shift the definition of the “digital sinc^3 filter” to the second page, where it is first mentioned. The form of the trajectory angle and of the offset equations should be improved.
I recommend this manuscript for publication in your journal.
Author Response
Thanks for the nice comments.
Regarding the shift of the definition of the “digital sinc^3 filter” to the second page I have modified slightly Section 2: Architecture of the Electronic Circuitry to include such a description of the sinc filter. This section has been reworded as follows:
The developed instrument comprises interfacing circuitry to a 3-axes Hall probe using the spinning current modulation technique [4 - 6] explained in detail in Section. 3 and further amplified in [13]. Each axis of the Hall probe is biased with a very high precision and temperature independent 2.5 mA current source. As the Hall probe has an on-die PT100 sensor, this is also interfaced to the instrument using the four-wire configuration so that lead wire resistance does not affect the true voltage temperature readout. In order to minimize any self-heating effects of the PT100 the bias current used is just 250 µA.
The four analogue differential voltages are all amplified, demodulated and low pass filtered to a 500 Hz bandwidth. This is performed by a 3rd order low pass fully differential Butterworth filter on each channel which serves as an antialiasing filter for the ADC and the internal digital sinc3 filter of the ADC. The antialiasing filter is designed with a bandwidth of 500 Hz providing a ripple free response and no attenuation in the passband that covers the full frequency response of the Hall probe.
All the differential signal paths are length matched and routed parallel to each other to optimize CMRR. Also each pair of the differential tracks are routed separately to the other pairs with a copper pour ground area in between in order to minimize crosstalk.
The four channel simultaneous sampling Delta-Sigma analogue-to-digital converter [7] digitizes the four analogue signals with 24-bit resolution. Oversampling techniques implemented through the Delta-Sigma architecture of the ADC enables the differential analogue input voltage to be sampled at an effective frequency of 4.096 MHz from the delta-sigma modulator as shown in Figure 1. The modulator then converts the analogue input signal into a high-speed, pulse-wave representation. Further details of the implementation can be found in [13].
The third order sinc filter on each channel of the ADC works in the digital domain as data is supplied to the filter from the modulator at the rate of fMOD. A detailed phase analysis and magnitude analysis are presented in Section 6 that pertain to the frequency response of the sinc3 filter. The digital sinc3 filter still operates at the modulator sampling rate for the decimator to be able to reduce the digital signal’s output rate to the desired Nyquist frequency according to the output data rate. The decimating function works by accumulating and averaging together groups of 24-bit data. In this way the actual output data rate is decimated down in the kHz range.
-------------------
Regarding the improvement in the form of the trajectory angle and of the offset equations I have modified this part as follows:
The characterization of the individual periods of one undulator segment is performed using a 3-axes Hall probe which is moved longitudinally along the laser line. B(z) is measured along the undulator length. The trajectory angle φ given by Equation (1) and the offset x given by Equation (2) are calculated locally at every magnet period and are corrected by vertical adjustment of the keeper support and the horizontal adjustment of the pole [2].
|
(1) |
|
(2) |
For this purpose, a novel three axis teslameter has been developed that is interfaced to a SENIS type S Hall probe [3] for the high fidelity characterization of the new line of the Athos undulators.

Reviewer 2 Report
A new instrument to provide more accurate measurement of magnetic field was presented. The overall approach is technically sound. Detailed information was presented regarding the circuitry, current modulation technique, temperature compensation/calibration and performance validation. Repeatability performance is really good. Noise performance can be improved to cover larger bandwidth. The limitation of SD card data transfer rate should be improved. From an instrumentation perspective, the presented work integrated many functions and the validation appears to be very convincing.Author Response
Thanks for the nice comments.
Yes analyzing the noise performance of the instrumentation setup at higher bandwidths than 500 Hz is a very valid point and would have shed more light from a quantitative aspect on how the performance deteriorates at higher frequencies. However for this application the main users were not really interested in achieving higher bandwidths and were mostly interested in the best achievable resolution at a limited bandwidth of 500 Hz thus performing all of the experimentation and validation at these relatively low frequencies.
The main limitation in the SD card data transfer rate finds its source in the maximal 20 MHz SPI clock frequency that the micro controller handles on these selected GPIOs. All the 50 MHz switching capable GPIOs on the microcontroller are taken up by the SDRAM memory module. Thus this issue is mostly a hardware restriction when considering that the micro controller does not handle the SD protocol to communicate with the SD card but it only handles SPI based communication for this particular peripheral.